

# *Nostolepis* scale remains (stem Chondrichthyes) from the Lower Devonian of Qujing, Yunnan, China

Qiang Li[1,2,3], Xindong Cui[3,4], Plamen Stanislavov Andreev[1,3], Wenjin Zhao[3,4,5], Jianhua Wang[1], Lijian Peng[1] and Min Zhu[3,4,5]

[1] Research Center of Natural History and Culture, Qujing Normal University, Qujing, China
[2] Chongqing Institute of Geology and Mineral Resources, Chongqing, China
[3] Key CAS Laboratory of Vertebrate Evolution and Human Origins, Institute of Vertebrate Paleontology and Paleoanthropology, Chinese Academy of Sciences (CAS), Beijing, China
[4] University of Chinese Academy of Sciences, Beijing, China
[5] CAS Center for Excellence in Life and Paleoenvironment, Beijing, China

Corresponding author
Min Zhu, zhumin@ivpp.ac.cn

## ABSTRACT

Based initially on microfossils, *Nostolepis* is one of the first known 'acanthodians', which constitute a paraphyletic assemblage of plesiomorphic members of the total group Chondrichthyes. Its wide distribution has potential implications for stratigraphic comparisons worldwide. Six species of *Nostolepis* have been reported in China, including one species from the Xitun Formation (Lochkovian, Lower Devonian) of Qujing, eastern Yunnan. Acid preparation of rock samples from the Xitun Formation has yielded abundant acanthodian remains. Based on both morphological and histological examinations, here we identify five species of *Nostolepis*, including two new species. *N. qujingensis* sp. nov. is characterized by thin scales devoid of the neck anteriorly and the dentine tubules rarely present in the anterior part of the crown. *N. digitus* sp. nov. is characterized by parallel ridges on anterior and lateral margins of the crown, and the neck constricted and ornamented with pore openings. We extend the duration of *N. striata* in China from the Pridoli of Silurian (Yulungssu Formation) to the Lower Devonian in Qujing and report the first occurrences of *N. amplifica* and *N. consueta* in this region. This study increases the diversity of the Lower Devonian Xitun Fauna and provides a better understanding of the paleogeographic distribution of *Nostolepis*.

## INTRODUCTION

The Lower Devonian Xitun Formation exposed in Qujing, China has yielded a diverse early vertebrate fauna, whose macrofossils are represented by jawless galeaspids and jawed members such as 'placoderms' and osteichthyans (*Chang, 1982*; *Chang & Yu, 1984*; *Zhu, 1996*; *Zhu, Yu & Janvier, 1999*; *Zhu, Yu & Ahlberg, 2001*; *Zhu & Yu, 2002*; *Zhu et al., 2006*). Probably due to the micromeric (small scales or tesserae on the head, rather than macromeric, with large dermal bones) skeletal nature of conventionally-defined chondrichthyans and 'acanthodians', no cranial remains of the two groups have been

found in the Xitun Formation, although microfossils from the same bonebeds have been assigned to the two groups (*Wang, 1984*). Recent revisions on the early vertebrate phylogeny suggest that conventionally-defined chondrichthyans and 'acanthodians' form a clade, i.e., the total group Chondrichthyes, whose micromeric skeleton probably evolved from an ancestral macromeric condition possessed by 'placoderms' and osteichthyans (*Brazeau, 2009*; *Davis, Finarelli & Coates, 2012*; *Zhu et al., 2013*; *Giles, Friedman & Brazeau, 2015*; *Long et al., 2015*; *Burrow et al., 2016*). A better understanding of the chondrichthyan members from the Xitun Formation will provide a further glimpse of the diversity of jawed vertebrates in South China during the Early Devonian.

*Nostolepis* is a cosmopolitan acanthodian genus erected by *Pander (1856)* for scales. It was generally assigned to the family Climatiidae of the order Climatiiformes for the latter half of the twentieth century (*Berg, 1940*; *Gross, 1957*; *Gross, 1971*; *Denison, 1979*; *Tông-Dzuy, 1994*; *Burrow, 1997*; *Miller & Märss, 1999*; *Vergoossen, 2002a*; *Valiukevičius, 2003c*; *Wang, 2003*; *Vergoossen, 2004*). However, *Burrow, Lelièvre & Janjou (2006)* referred it to the Climatiidae order indet. Considering that *Nostolepis* lacks circumorbital bones, dermal pectoral girdle, and branchial plates, *Burrow & Turner (2010)* doubted its assignment to the Climatiidae. So far, the order and the family for *Nostolepis* have been uncertain (*Burrow & Murphy, 2016*).

*Nostolepis* was first reported in China as *Nostolepis* sp. indet. from the Xitun Formation (Lochkovian, Lower Devonian) in Qujing (*Wang, 1984*). The first nominal species of *Nostolepis* in China is *N. sinica* from the Yulungssu Formation (Pridoli, Silurian) in Qujing (*Gagnier, Jahnke & Shi, 1989*). Soon after, *N. striata* and *N.* sp. were described from the same horizon in Qujing (*Wang & Dong, 1989*). *N. striata* was also recorded from the Xiaputonggou Formation (Lochkovian) in West Qinling (*Wang et al., 1998*) and the Shanjiang Formation (Homerian, Wenlock) in Lijiang, Yunnan (*Wang, 2003*). *N. gracilis* was described from the Alengchu Formation and Shanjiang Formation (Wenlock, Silurian to Emsian, Lower Devonian) in Lijiang, Yunnan (*Wang, 2003*) and the Xiaputonggou Formation (Lochkovian) in West Qinling (*Wang et al., 1998*). From the latter site, *Wang et al. (1998)* also described *N. tewonensis* from Homerian (Wenlock, Silurian) to Lochkovian. *Nostolepis guangxiensis* was reassigned to *Nostovicina guangxiensis* because its crown lacks the Stranggewebe (*Valiukevičius & Burrow, 2005*).

Here, we describe five species of *Nostolepis*, including two new species, from the Xitun Formation in Qujing, Yunnan, China, thus increasing the diversity of the Early Devonian Xitun Fauna. We also summarize and discuss the paleogeographic distribution and stratigraphical ranges of *Nostolepis* species.

## MATERIALS & METHODS

This study is based on isolated scales of *Nostolepis* from the Xitun Formation (Lochkovian, Lower Devonian) in Xitun, Xishan subdistrict, Qujing, Yunnan, China (N:25°31.547′, E:103°31.547′). Studied scales were extracted by treatment with buffered 5% acetic acid from greenish-yellow muddy limestone of the Xitun Formation at the laboratory of Qujing Normal University.

Thin sections were prepared by embedding scales in epoxy resin. The cured resin blocks were cut with a low speed saw and 100–200 mm away from embedded specimens. Then the surface was ground with sand paper with grit sizes ranging from P600 to P4000 until the desired surface of the specimen was exposed. This surface was polished with a grinder/polisher and glued to a glass slide. Finally, the other surface was cut, ground and polished in the same way to produce doubly-polished thin sections. Scales of each species were sectioned longitudinally, transversely and horizontally. Scanning electron microscope (Hitachi S-3700N) was used to take images of intact scales. All specimens are housed in the collection of the Institute of Vertebrate Paleontology and Paleoanthropology (IVPP), China.

The electronic version of this article in Portable Document Format (PDF) will represent a published work according to the International Commission on Zoological Nomenclature (ICZN), and hence the new names contained in the electronic version are effectively published under that Code from the electronic edition alone. This published work and the nomenclatural acts it contains have been registered in ZooBank, the online registration system for the ICZN. The ZooBank LSIDs (Life Science Identifiers) can be resolved and the associated information viewed through any standard web browser by appending the LSID to the prefix http://zoobank.org/. The LSID for this publication is: urn:lsid:zoobank.org:pub:C3957E52-DD5E-438E-BCD2-515B1611D9C2. The online version of this work is archived and available from the following digital repositories: PeerJ, PubMed Central and CLOCKSS.

# RESULTS

## Systematic paleontology

Referred Chinese Material: 69 trunk scales (IVPP V26830.1–V26830.68, V26831).

Description:

Morphotype 1 (Figs. 1A–1D):

This type of scales has a broad triangular, flat, or slightly inclined crown. Crowns are about 0.21–0.56 mm long and 0.28–0.51 mm wide. Three to four short, stout, longitudinal and converging ridges bend inward from the anterior face of the crown to a quarter of the crown's length. The distal pointed end of the crown overhangs the base slightly. Most of the scales have two smooth lateral slopes rising from the anterior crown-neck boundary and joining at the pointed posterior apex. The neck is smooth and indistinctive. The rhomboid base is strongly or moderately convex and its outline matches that of the crown except for the crown's posterior end.

Morphotype 2 (Figs. 1E–1H):

One triangular scale form has an inclined crown surface with an angle of nearly 30°. The crowns are slightly wider than long (0.5 mm wide and 0.3 mm long). Two strong ridges are running back from the anterior crown margin to one-third of the crown length with 1–3 subparallel ridges between them. The ridges slope down almost to the base rim anteriorly, thus distinctly lowering the scale's anterior neck. The lateral slopes of the crown

'Acanthodii' (*Owen, 1846*)
Order and Family Incertae sedis
Genus *Nostolepis Pander, 1856*
*Nostolepis striata Pander, 1856*

1856 *Nostolepis striata*, Pander, pl. 28, figs. 1, 7.
1984 *Nostolepis* sp., Blieck et al., pl. 1, figs. 3–4.
1997 *Nostolepis striata*, Burrow, figs. 2A–B; pl. 1, figs. 13A–C; pl. 2, figs. 1–4.
1998 *Nostolepis striata*, Valiukevičius, pl. 1, figs. 1–4; pl. 2, figs. 4–7.
1998 *Nostolepis striata*, Wang et al., pl. 1, figs. B–C.
1999 *Nostolepis striata*, Vergoossen, pl. 2, fig. 15.
2002b *Nostolepis striata*, Vergoossen, pl. 3, figs. 26–36.
2003a *Nostolepis striata*, Vergoossen, pl. 6, figs. 71–76.
2003b *Nostolepis striata*, Vergoossen, pl. 10, figs. 85–100.
2003 *Nostolepis striata*, Wang, fig. 1.
2004 *Nostolepis striata*, Vergoossen, pl. 5, figs. 54–61.
2005 *Nostolepis striata*, Valiukevičius, fig. 3L and M.
2013 *Nostolepis striata*, Burrow et al., figs. 3, 4.1 and 4.2.
2018 *Nostolepis striata*, Turner and Burrow, figs. 6A–D.

Other synonyms see (*Valiukevičius, 1998*).

are sharp and well developed. The neck is short, smooth anteriorly, deeper and more concave posteriorly. The base is rhomboid and moderately convex.

Histology (Figs. 2A–2B):

The scales of the first and second varieties have the same histology. The crown is composed of 3–4 odontodes, which are thick on top and sides. All the odontodes are filled with a network of dentine tubules. The canal system is visible on the crown's posterior part, but Stranggewebe only present in the primordial odontode. The base is pyramid-shaped and is composed of cellular bone. Long Sharpey's fibers penetrate the base radially.

Remarks:

*Nostolepis striata* has been frequently redefined since it was erected by *Pander (1856)* for scales from Estonia. *Gross (1947)* reassigned some acanthodian species, erected by *Brotzen (1934)*, to *Nostolepis striata*. *Vergoossen (2002b)* agreed with this assignment, but thought the morphological variations of *N. striata* were still too large. After careful observation, *Vergoossen (2002b)* defined seven different morphotypes according to the *Nostolepis* scales from Klinta, southern Sweden. We follow his classification and recent descriptions by *Burrow et al. (2013)* and *Turner & Burrow (2018)*. We suggest two kinds of morphotypes here based on the new materials from the Xitun Formation. We combine morphotypes 1, 2 and 6, defined in *Vergoossen (2002b)*, within one group, and morphotypes 3 and 7 within another group. This second kind of scales comprises

pinnal scales or tesserae as in *Turner & Burrow (2018)*. Both morphotypes have the typical *Nostolepis* structure with mesodentine and Stranggewebe.

*Nostolepis amplifica Valiukevičius, 2003c*

1984 *Nostolepis* sp. indet., Wang, figs. 3D–F.
1989 *Nostolepis sinica*, Gagnier et al., pl. 1, figs. 3–4?
1999 *Nostolepis striata*, Vergoossen, pl. 2, figs. 16–17.
2003c *Nostolepis amplifica*, Valiukevičius, figs. 2A–H and 3.
2005 *Nostolepis amplifica*, Valiukevičius, figs. 2A–B.

Referred Chinese Material: 43 trunk scales (IVPP V7216.4, IVPP V26832.1–V26832.41, IVPP V26833).

Description:

Morphotype 1 (Figs. 1I–1N):

The scale crown's length and width range from 0.32–0.66 mm and 0.31–0.65 mm, respectively. Morphotype 1 is elongated with rhombic to ellipsoidal in outline. The crown surface is side to side and inclined, with one-fourth of the crown's length overhanging the base posteriorly. Two longest prominent ridges from an elevated median area of the crown, are sculptured by 2–6 rounded, short, subparallel anterior ridges. Most scales have an unsculptured lateral slope on each side of the median area, but some have several ridgelets (Figs. 1M–1N), which are finer and lower than those on the median area. The neck is well defined and constricted. A row of 3–5 regularly spaced small openings are located on each side of the neck, and sometimes they are also visible anteriorly below the ridges (Figs. 1I–1L). The convex and rectangular base is equal or slightly larger than the crown.

Morphotype 2 (Figs. 1O–1P):

These scales have an inclined median crown surface with two strong main ridges and two inconspicuous lateral slopes on each side of the median area. Crown length varies from 0.60–0.75 mm and width from 0.50–0.65 mm. A shallow and wide longitudinal groove between the two central ridges is another characteristic, forming an incised anterior crown margin. The neck has a row of regularly spaced small openings locating on the side or below the ridge. The scale base of both morphotypes is shaped from isometric to asymmetric rhomboid, sometimes wider than long. The base protrudes anteriorly beyond the crown. It is moderately deep to deep, with the deepest point locating centrally or anteriorly.

Histology (Figs. 2C–2D):

The structures of tissues are similar in the two types. The crown is composed of mesodentine but the stranglacunae are not visible due to the recrystallization. The winding and short dentine tubules orient upwards. Ascending and vaulted vascular canals can be seen in the posterior part of the crown. Stranggewebe is short, narrow, and medium dense, developing only in the first and second lamellae on the crown's posterior part. The base consists of cellular bone with osteocytes.

Remarks:

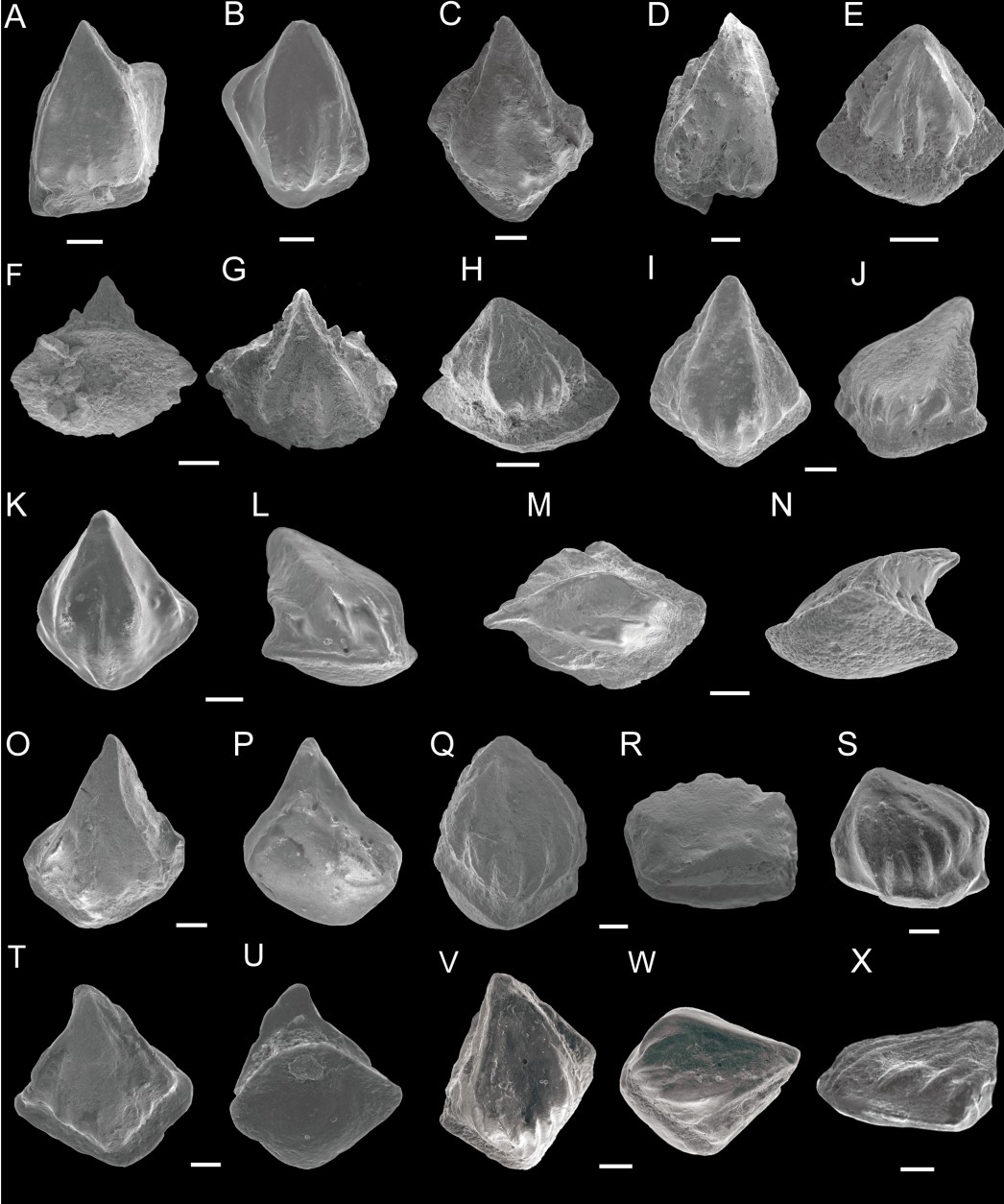

**Figure 1** **SEM photos of *Nostolepis striata*, *Nostolepis amplifica* and *Nostolepis consueta* scales.** (A)–(H) *Nostolepis striata*. (A) Crown view, IVPP V26830.1. (B) Crown view, IVPP V26830.2. (C) Crown view, IVPP V26830.3. (D) Crown view, IVPP V26830.4. (E) Crown view, IVPP V26830.5. (F) Base view and (G) Crown view, IVPP V26830.6. (H) Crown view, IVPP V26830.7. (I)–(P) *Nostolepis amplifica*. (I) Crown view and (J) Antero-lateral view, IVPP V26832.1. (K) Crown view and (L) Antero-lateral view, IVPP V26832.2. (M) Crown view and (N) Lateral view, IVPP V26832.3. (O) Crown view and (P) Base view, IVPP V26832.4. (Q)–(X) *Nostolepis consueta*. (Q) Crown view and (R) Postero-lateral view, IVPP V26834.1. (S) Antero-lateral view, IVPP V26834.2. (T) Crown view and (U) Base view, IVPP V26834.3. (V) Crown view and (W) Antero-lateral view, IVPP V26834.4. (X) Antero-lateral view, IVPP V26834.5. Scale bars 0.1 mm. Early Devonian, Lochkovian, the Xitun Formation, Qujing.

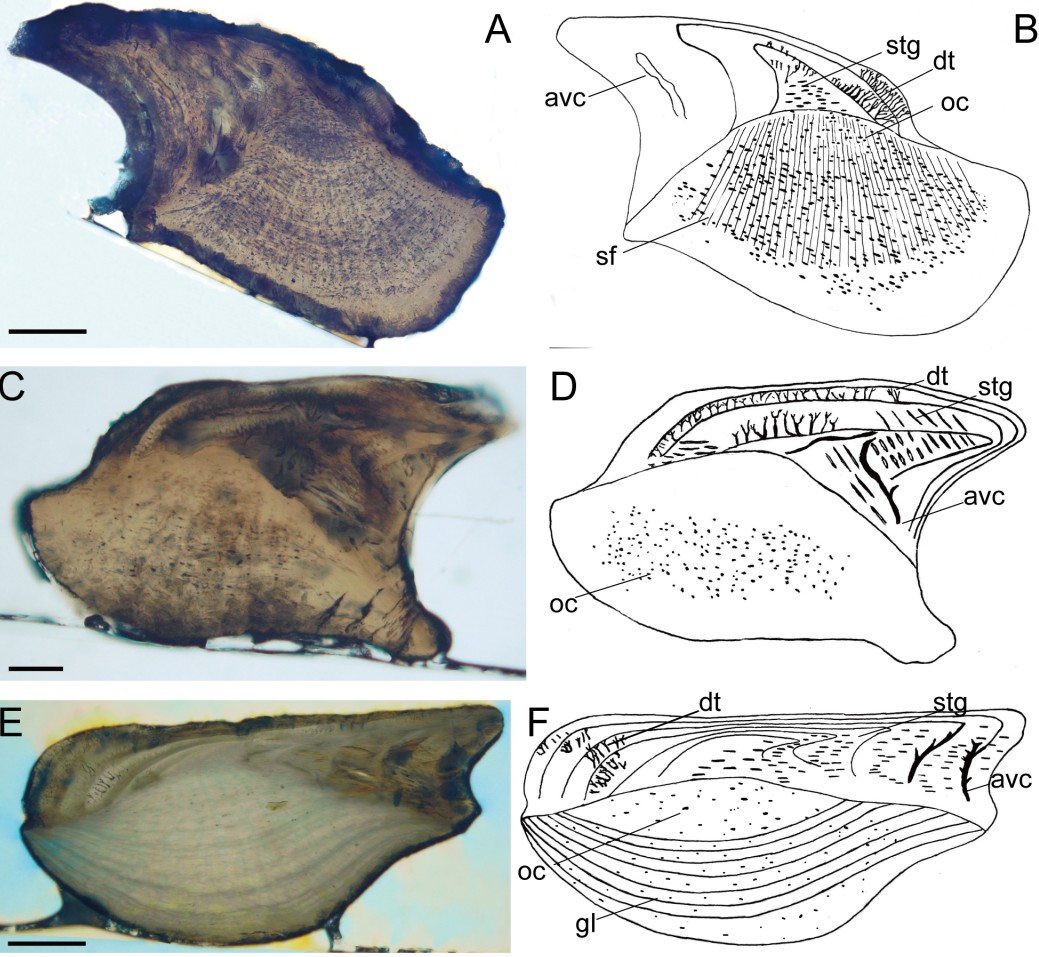

**Figure 2** **Histological microstructure and illustrative drawings of *Nostolepis striata, Nostolepis amplifica*, and *Nostolepis consueta* scales in vertical longitudinal sections.** (A)–(B) *Nostolepis striata* IVPP, V26831. (C)–(D) *Nostolepis amplifica* IVPP, V26833. (E)–(F) *Nostolepis consueta* IVPP, V26835. *dt*, dentine tubule; *oc*, osteocyte cavity; *stg, Stranggewebe, gl*, growth lamella; *avc*, ascending vascular canal; *sf*, Sharpey's fibers. Scale bars 0.1 mm.

This species was erected by *Valiukevičius (2003c)* for scales from Pridoli and Ludlow of Lithuania. *Valiukevičius (2003c)* indicated that *N. amplifica* was similar to the specimens of *N. striata* described by *Vergoossen (2002b)* (pl. 14, figs. 85–87) obtained from Öved-Ramsåsa, the Sandstone Formation, Ludlow to Pridoli and (*Vergoossen, 2000*) pl. 1, figs 4–5 and 7–8), obtained from Welsh Borderland, Silurian to Devonian. We consider that the specimens described by *Vergoossen (2002b)* (pl. 14, figs 85–87) should be assigned to *N. striata* as originally suggested. Morphologically, *N. amplifica* is similar to *N. striata*, and the former usually has a shallow and longitudinal groove between the two central ridges. Another difference is that the neck has a row of regularly spaced small openings for *N. amplifica*, but it is rarely reported for *N. striata*. Only two exceptional specimens with neck openings were reported in *N. striata*, which differ from the trunk scales described by *Gross*

*(1947)*, pl. 7, fig. 13 and *Gross (1971)*, pl. 5, fig 3a. Histologically, *Valiukevičius (2003c)* also figured that both species have similar mesodentine, Stranggewebe, vascular canals, and large number of osteocytes, but the canal system is more advanced and Stranggewebe is shorter and denser in *N. amplifica*. *N. amplifica* was first found by *Wang (1984)* (figs. 3D–3F) and described as *Nostolepis* sp. indet. from the Xitun Formation. In terms of morphology, some specimens described by *Gagnier, Jahnke & Shi (1989)* (pl. 1, figs 3–4) from the Yulungssu Formation, Hongmiao (Shiyanpo), Qujing, which differ from the holotype of *N. sinica* (pl. 1, figs 1–2), probably belong to *N. amplifica*.

*Nostolepis consueta* *Valiukevičius, 2003c*

1984 *Nostolepis* sp. indet., Wang, figs. 3A–C.
1997 *Gomphonchus sandelensis*? Märss, pl. 3, fig. 5.
1998 *Nostolepis minima*, Valiukevičius, pl. 1, fig. 5.
1998 *Nostolepis* sp. or *Cheiracanthoides* sp., Valiukevičius, pl. 1. figs. 10–15.
1998 *Nostolepis striata*, Wang et al., pl. 1, fig. A.?
2003c *Nostolepis consueta*, Valiukevičius, figs. 5A–I, 6 and 7.
2005 *Nostolepis consueta*, Valiukevičius, figs. 2E–F.

Emended diagnosis: The crowns are flat, never inclined, and sculptured with 6–12 short and parallel ridges. Low anterior ridges fade out at one-third of the crown's length. The posterior part of the crown is composed of Stranggewebe and widened ascending vascular canals. The Stranggewebe is dense in each odontode.
Referred Chinese Material: around 25 trunk scales (IVPP V7216.7, IVPP V26834.1–V26834.23, V26835).
Description (Figs. 1Q–1X):
  The scale crown is flat, never inclined, and has a nearly triangular or ovoid shape. Crown length varies from 0.37–0.70 mm and width from 0.41–0.55 mm, respectively. Most of the scale crowns have 3–12 subparallel and short ridges extending one–third of the crown's length, converging towards the posterior end. Lateral slopes are not well developed in most of the scales. The anterior edge of the crown is convex, curving down to the base. The neck is smooth and constricted for most of the scales. A rhombic base is slightly convex, anteriorly vaulted, so that the base protrudes in front of the crown.
Histology (Figs. 2E–2F):
  The crown has a relatively big primordial scale and a few growth lamellae thin on top but thick on anterior and posterior parts. The odontodes have dentine tubules, which are dense anteriorly. The crown's posterior part is composed of Stranggewebe and widened ascending and vaulted vascular canals (over the base). Stranggewebe is short and densely presented in each odontode. The base is composed of compact, thin-lamellar bone, containing a large number of osteocytes.
Remarks:
  For trunk scales, *Valiukevičius (2003c)* identified two morphotypes of *N. consueta*. The first morphotype has 8–12 ridges, and the second morphotype has fewer than 6 ridges. In

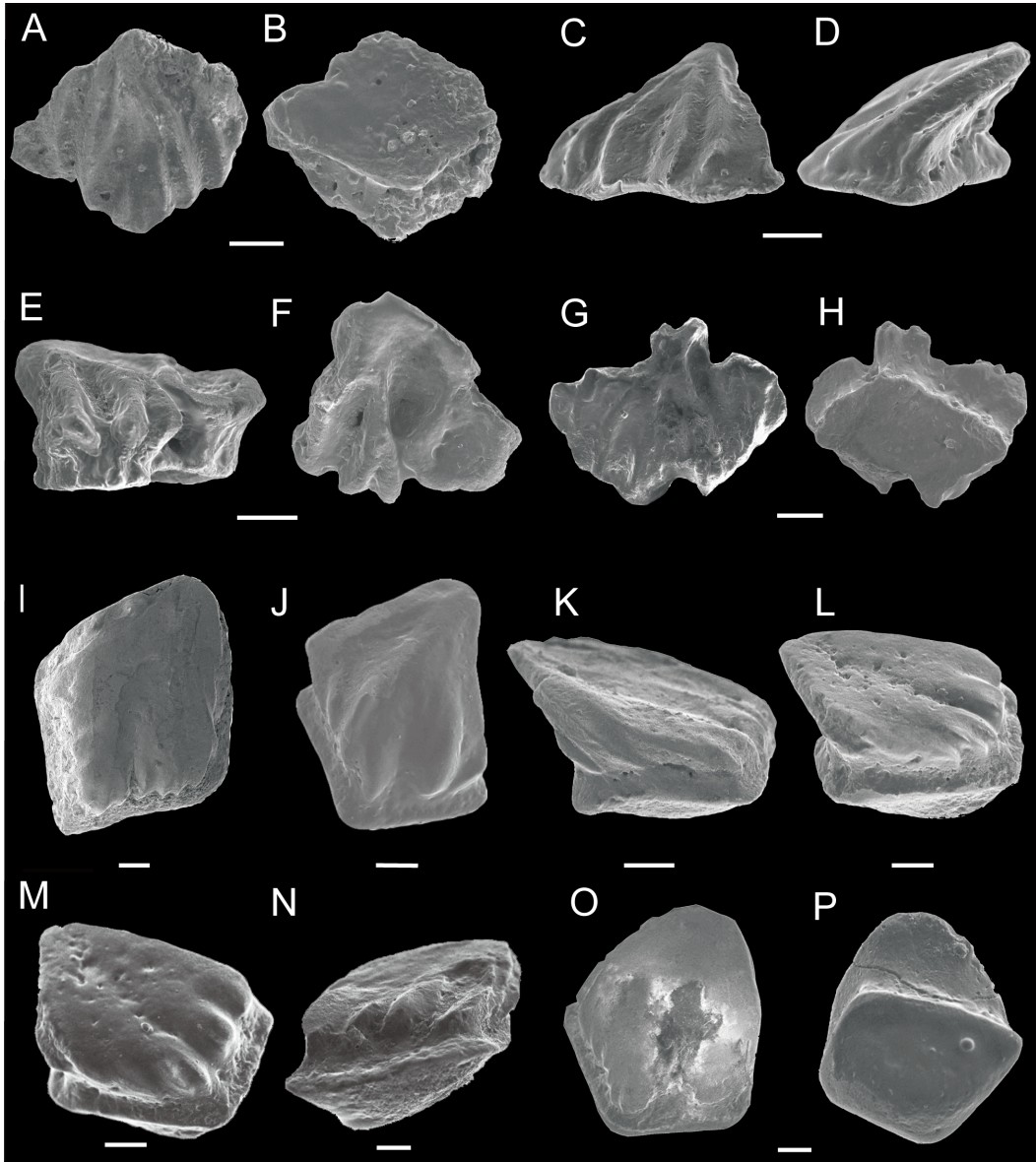

**Figure 3** **SEM photos of *Nostolepis qujingensis* sp. nov. and *Nostolepis digitus* sp. nov. scales.** (A)–(H) *Nostolepis qujingensis* sp. nov . (A) Crow view and (B) Base view, IVPP V26838.1, holotype. (C) Crown view and (D) Lateral view, IVPP V26838.2. (E) Antero-lateral view and (F) Crown view, IVPP V26838.3. (G) Crown view and (H) Base view, IVPP V26838.4. (I)–(P) *Nostolepis digitus* sp. nov. (I) Crown view, IVPP V26840.1, holotype. (J) Crown view, IVPP V26840.2. (K) Antero-lateral view, IVPP V26840.3. (L) Antero-lateral view, IVPP V26840.4. (M) Antero-lateral view, IVPP V26840.5. (N) Lateral view, IVPP V26840.6. (O) Crown view and (P) Base view, IVPP V26840.7. Scale bars 0.1 mm. Early Devonian, Lochkovian, Xitun Formation, Qujing.

our collection, most of the referred scales have 6–10 ridges. *N. consueta* is so similar to *N. striata* that some *N. consueta* probably have been misassigned to the latter. *N. consueta* has a flat crown, never inclined, and the anterior corner of the crown is markedly angular. More differences are present in the histological features. *N. consueta* has dense Stranggewebe in

each odontode; *N. consueta* has a low number of osteocytes and no obvious Sharpey's fibers in the base compared with *N. striata*. The scales assigned to *Nostolepis* sp. indet. (*Wang, 1984*), figs. 3A–3C) should be *Nostolepis striata*. And the specimen (V11528.1) described as *Nostolepis striata* (*Wang et al., 1998*, pl. 1, fig. A) from the lower Lochkovian Xiaputonggou Formation, West Qinling, China, probably belong to *N. consueta*.

*Nostolepis qujingensis* sp. nov.

Deviation of name: After Qujing city, the fossil site.
Holotype: IVPP V26838.1.
Type locality and horizon: Xitun, Xishan subdistrict, Qujing, Yunnan, China; Xitun Formation, Lochkovian, Lower Devonian.
Referred Material: 19 trunk scales (IVPP V26838.1–V26838.16, IVPP V26839-V26839.3)
Diagnosis:

Trunk scales are of moderate size, ornamented with longitudinal ridges in the crown. Crown length is 0.3–0.5 mm, width 0.3–0.5 mm. The distal pointed end of the crown always overhangs the base at one-third of its length. The angle between crown and base can reach 45°. The neck is absent anteriorly, but deeper and concave posteriorly. The base is rhomboid, thin, and concave. Dentine tubules rarely present in the anterior part of the crown. Scale base is composed of bone with few osteocytes.

Description (Figs. 3A–3H):

The scale crowns are triangular in shape or irregularly shaped (Figs. 3E–3H). They incline antero-posteriorly at an angle of 30°–45°. Crown length varies from 0.33–0.50 mm and width from 0.36–0.51 mm, respectively. The crown is sculptured with slightly sinuous ridges extending the whole surface (Figs. 3A–3D) or ornamented with radial ridges and deep grooves (Figs. 3E–3H). The lateral slope usually develops in the former morphotype (Figs. 3A–3D). The neck is absent anteriorly, but deeper and concave posteriorly. The base is rhomboid and thin, never protrudes beyond the crown on all sides.

Histology (Figs. 4A–4D):

The crown's odontodes are very thin, and the dentine tubules are rarely present in each odontode. Wide vascular canals rise from the neck and penetrate the posterior part of the crown. Scattered osteocytes form a low and flat base.

Remarks:

*Nostolepis qujingensis* is similar to *N. lineleyensis* in the crown sculptured with ridges and grooves at the anterior margin, sloping down at an angle of 30°–45°. But *N. lineleyensis* has a much thicker crown and more ridges than *N. qujingensis*. More differences are found in histology. *N. qujingensis* and *N. lineleyensis* have a similar system of vascular canals, but the
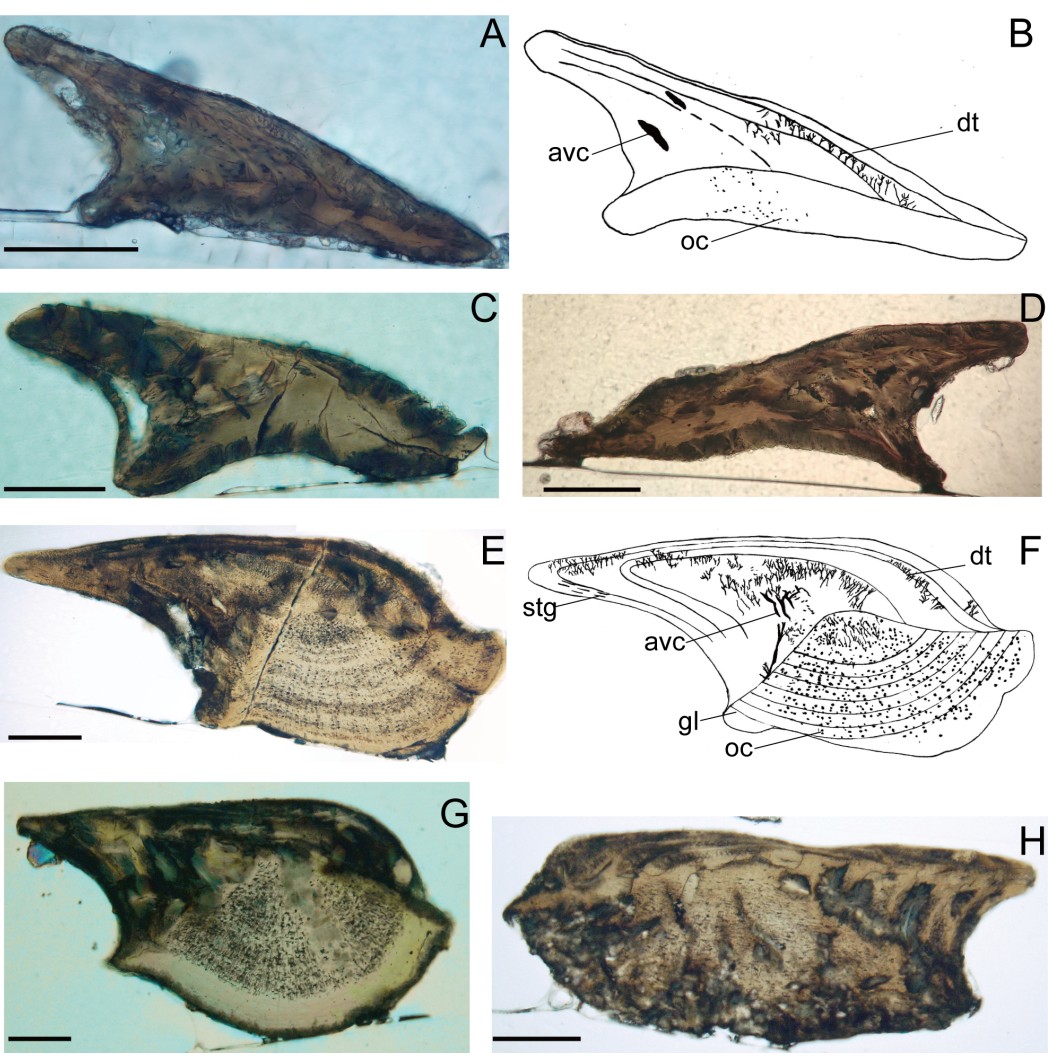

**Figure 4** Histological microstructure and illustrative drawings of *Nostolepis qujingensis* sp. nov. and *Nostolepis digitus* sp. nov. scales in vertical longitudinal sections. (A)–(D) *Nostolepis qujingensis* sp. nov. (A)–(B) IVPP V26839. (C) IVPP V26839.2. (D) IVPP V26839.3 (E)–(H) IVPP *Nostolepis digitus* sp. nov. (E)–(F) IVPP V26841. (G) IVPP V26841.2. (H) IVPP V26841.3. *dt*, dentine tubule; *oc*, osteocyte cavity; *stg*, *Stranggewebe, gl* growth lamella; *avc*, ascending vascular canal; *sf*, Sharpey's fibers. Scale bars 0.1 mm.

latter has a typical mesodentine in the crown and cellular bone with numerous large and elongate osteocytes (*Miller & Märss, 1999*).

*Nostolepis digitus* sp. nov.

Derivation of name: From the Latin *digitus*, referring to the appearance of the crown ornamentation.
Holotype: IVPP V26840.1
Type locality and horizon: Xitun, Xishan subdistrict, Qujing, Yunnan, China; Xitun Formation, Lochkovian, Lower Devonian.

Referred Material: 28 trunk scales (IVPP V26840.1–V26840.25, IVPP V26841-IVPP V26841.3).

Diagnosis:

Large-sized scale having a sub-rhomboid crown with rounded and widened anterior edge and a slightly tapered posterior end. Short and parallel ridges on the crown's anterior and lateral margins, curving down to the base. The neck is constricted with small pores. Scale bases are strongly convex; their maximum depth is slightly in front of the center. Dentine tubules are dense in each odontode of the crown, and the Stranggewebe is oriented parallel to the growth zone lines and distributed in the posterior part of the odontodes. Ascending vascular canals are well developed. The base is filled with numerous osteocytes.

Description (Figs. 3I–3P):

Scale crowns are usually flat, or slightly inclined. Crown length varies from 0.45–0.76 mm and width from 0.41–0.70 mm. The rhombic crown has a round and wide anterior edge and a blunt posterior cusp. The posterior parts of crowns tilt to one side. The crown is ornamented with short and parallel ridges that do not extend to the middle of the crown posteriorly, but bend down anteriorly and laterally. The crown has a smooth lateral slope flanking the posterior part. The neck is short, lowered anteriorly, but deeper and concave posteriorly. Only small pores are visible on the neck. The base is strongly convex, vaulted anteriorly and protruding in front of the crown.

Histology (Figs. 4E–4H):

The crown is composed of 3–4 odontodes, and the young one envelops the older one completely. Dentine tubules are dense and discernable in each odontode. The growth lamella contains long ascending vascular canals placed posteriorly in the neck over the scale base. The Stranggewebe is oriented parallel to the growth zone lines and only presented in the posterior part of the outermost odontode. The upper surface of the base forms a high pyramid, and the base is filled with many osteocyte cavities.

Remarks:

The referred specimens are similar to *Nostolepoides mingyinensis* in morphology and histology, which was described from another Devonian site in Yunnan Province (*Wang, 2003*) (p. 8, fig. 3). Morphologically, the crowns of both species are rhomboid, sculptured with short and parallel ridges, not extending half of the crown length, and the lateral slopes are less developed. *N. minyingensis* has very sharp ridges and anterior crown margin, and *N. digitus* has short rounded ridges and crown margin. Histologically, their base tissues are similar in having dense osteocytes. The Stranggewebe is not visible in *Nostolepoides mingyinensis*, and the crown is composed of more odontodes. Morphologically, *N. digitus* is also similar to *Nostolepis* sp. aff. *N. multicostata* reported by *Burrow, Lelièvre & Janjou (2006)* (figs. 3, 11–12 and 15–16) from the Lower Devonian Jawf Formation of northwestern Saudi Arabia. But the Arabian form is much larger than *N. digitus*, and its crown has more ridges.

## DISCUSSION AND CONCLUSION

*Valiukevičius (2005)* classified the acanthodian scales with *Nostolepis*-like histology into five groups based on the presence or absence and extent of the Stranggewebe, and different

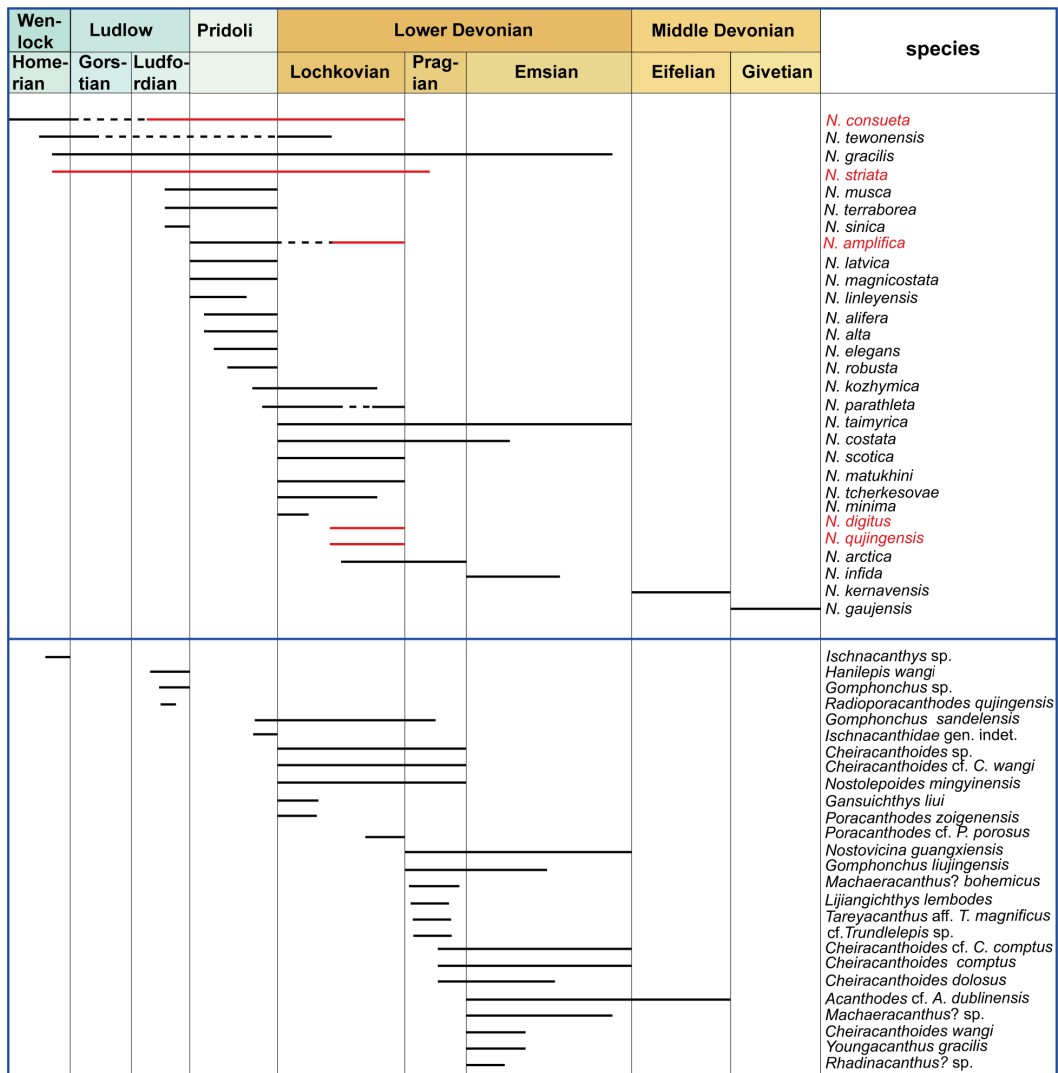

**Figure 5  Stratigraphical ranges of *Nostolepis* in the world and acanthodians in China from Silurian to Middle Devonian.** Modified from *Wang (2003)*, *Valiukevičius (2003b)*; *Valiukevičius (2005)* and *Zhao & Zhu (2015)*.

types of dentine tubules in scale crowns, which we follow here. Only the scales composed of typical Stranggewebe and simple odontocytic mesodentine in the crowns are referred to *Nostolepis*, which is in accordance with the classical diagnosis of *Nostolepis* (*Pander, 1856*; *Gross, 1971*).

The new *Nostolepis* microfossils in this study indicate a high diversity of 'acanthodian' taxa in the Xitun Formation. So far, 34 acanthodian species have been described from the Silurian and Devonian of China, including eight species of *Nostolepis* in total (Fig. 5). The other 21 *Nostolepis* species from the rest of the world are also listed here (Fig. 5). Here, we focus on the biostratigraphic and paleogeographic distribution of the *Nostolepis* species from China and discuss their potential biogeographic implications.

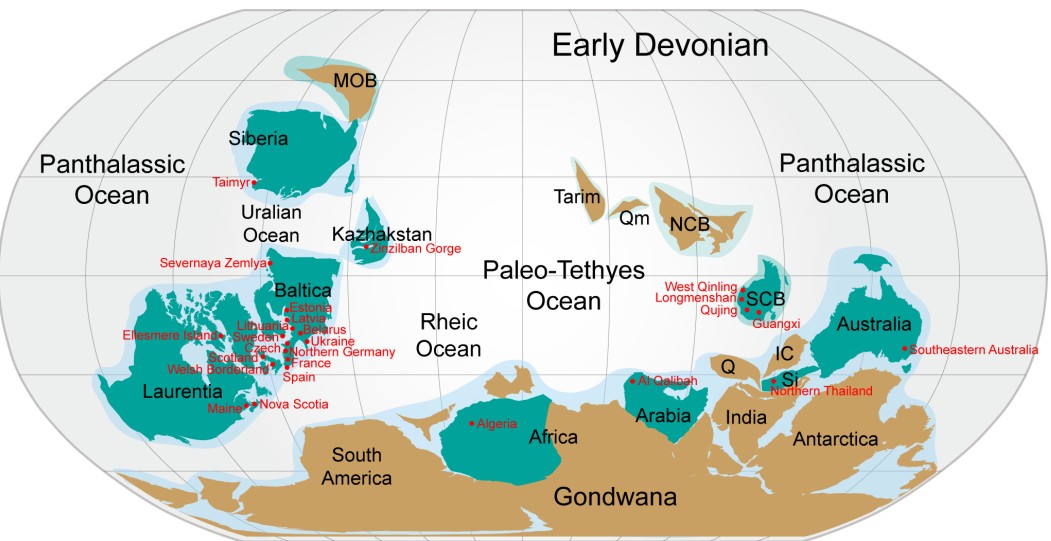

**Figure 6** ***Nostolepis* distribution in Lower Palaeozoic terranes around 400 Ma (Early Devonian).** Modified from *Huang et al. (2018)*.

As shown in Fig. 5, most *Nostolepis* species have been recorded from the Pridoli to Lochkovian, except for *N. infida* that was recorded in the Emsian. *N. musca* and *N. terraborea* were found in the beds from the Ludlow to Pridoli, and the earliest records of *N. consueta, N. tewonensis, N. gracilis* and *N. striata* are back to the Wenlock (*Denison, 1979*; *Gagnier, Jahnke & Shi, 1989*; *Wang & Dong, 1989*; *Wang et al., 1998*; *Wang, 2003*; *Valiukevičius, 2003a*; *Valiukevičius, 2004a*; *Valiukevičius, 2005*; *Burrow, 2013*; *Turner & Burrow, 2018*; *Wang et al., 2018*). *N. kernavensis* and *N. gaujensis* are the only taxa from the Middle Devonian and Upper Devonian (*Valiukevičius, 1985*; *Burrow, Janvier & Villarroel, 2003*; *Plax, 2011*; *Pinakhina & Märss, 2018*). *N. striata* from many Silurian sites illustrates that it migrated widely (*Pander, 1856*; *Fredholm, 1988*; *Märss, 1997*; *Burrow, 2013*). *N.consueta* and *N. amplifica* might have migrated from the Baltica Block to South China (*Valiukevičius, 2003c*).

Some widely distributed 'acanthodians' such as *Gomphonchus*, Ischnacanthidae, *Cheiracanthoides, Radioporacanthodes, Acanthodes* and *Machaeracanthus* have also been found in China, as shown in Fig. 5 (*Wang & Dong, 1989*; *Gagnier, Jahnke & Shi, 1989*; *Wang, 1992*; *Wang et al., 1998*; *Burrow, Turner & Wang, 2000*; *Wang, 2003*). But *Nostolepoides mingyinensis, Gansuichthys liui,* and *Lijiangichthys lembodes* were endemic to China (*Wang et al., 1998*; *Wang, 2003*). Most of these 'acanthodians' were described from the Devonian. But *Ischnacanthus* sp., *Hanilepis wangi, Gomphonchus* sp., *Radioporacanthodes qujingensis, Gomphonchus sandelensis* and Ischnacanthidae gen. indet. were reported from the Silurian (*Wang & Dong, 1989*; *Wang et al., 1998*) and *Acanthodes* cf. *A. dublinensis* extends to the Eifelian (*Burrow, Turner & Wang, 2000*).

Almost all the known *Nostolepis* species were distributed around the Paleo–Tethys Ocean (Fig. 6). *N. consueta, N. striata* and *N. amplifica* described here (Xitun Formation, Lochkovian) were all recorded from the Baltica Block (*Valiukevičius, 2003c*; *Valiukevičius,*

*2005*). Our new data of *Nostolepis* from Qujing suggest a connection with East Baltica. *N. striata* is a cosmopolitan species, also described from the Laurentia and Australia blocks (*Babin et al. 1976*; *Denison, 1979*; *Mader, 1986*; *Valiukevičius, 2005*; *Burrow et al., 2013*; *Plax, 2015*; *Turner & Burrow, 2018*). Some other *Nostolepis* species, i.e., *N. alifera*, *N. alta*, *N. elegans*, *N. costata*, *N. robusta*, *N. magnicostata*, *N. minima*, *N. gaujensis*, *N. latvica*, *N. kozhymica*, *N. parathleta*, *N. taimyrica*, *N. terraborea* and *N. linleyensis* were also recorded in the Baltica Block, but none of them has been reported in China (*Valiukevičius, 1994*; *Miller & Märss, 1999*; *Valiukevičius, 2003b*; *Valiukevičius, 2005*; *Turner et al., 2017*; *Pinakhina & Märss, 2018*). *N. infida*, *N. arctica*, *N. matukhini*, *N. minima* and *N. taimyrica* were also found in Siberia blocks (*Valiukevičius, 1994*). *N. costata* scales was recorded in the Arabia and some *Nostolepis* spp. were recorded in the Kazhakstan, Africa and Sibumasu blocks (Blieck et al., 1984, *Tông-Dzuy & Janvier, 1994*; *Tông-Dzuy & Janvier, 1990*; *Burrow, Lelièvre & Janjou, 2006*; *Burrow, Ivanov & Rodina, 2010*; *Mergal, Vaškaninová & Žigaite Ž, 2017*).

In summary, most of the *Nostolepis* species range from the Upper Silurian to Lower Devonian. *Nostolepis* is distributed mainly in the South China, Baltica, Siberia, Laurentia, and Australia blocks, with rare records in the Arabia, Africa, Sibumasu and Kazhakstan blocks. Biostratigraphic data of *Nostolepis* were very similar between the South China and Baltica blocks.

## ACKNOWLEDGEMENTS

We thank Qingming Qu (School of Life Sciences, Xiamen University) for providing constructive comments and suggestions that led to the improvement of our manuscript. We also thank Baochun Huang (Peking University) for providing the base map of Figure 6, Liantao Jia (IVPP) for the assistance in taking photographs, Yemao Hou (IVPP) and Pengfei Yin (IVPP) helped with CT scanning of some specimens.

### Funding

This research was supported by the Strategic Priority Research Program of the Chinese Academy of Sciences (XDA19050102, XDB26000000), the National Natural Science Foundation of China (41530102), the Key Research Program of Frontier Sciences, CAS (QYZDJ-SSW-DQC002), and the Chinese Postdoctoral Science Foundation grant (2019M663440). The funders had no role in study design, data collection and analysis, decision to publish, or preparation of the manuscript.

### Grant Disclosures

The following grant information was disclosed by the authors:
Strategic Priority Research Program of the Chinese Academy of Sciences: XDA19050102, XDB26000000.
National Natural Science Foundation of China: 41530102.

Key Research Program of Frontier Sciences, CAS: QYZDJ-SSW-DQC002.
Chinese Postdoctoral Science Foundation: 2019M663440.

## Competing Interests

The authors declare there are no competing interests.

## Author Contributions

- Qiang Li performed the experiments, analyzed the data, prepared figures and/or tables, authored or reviewed drafts of the paper, and approved the final draft.
- Xindong Cui analyzed the data, prepared figures and/or tables, authored or reviewed drafts of the paper, and approved the final draft.
- Plamen Stanislavov Andreev performed the experiments, analyzed the data, prepared figures and/or tables, and approved the final draft.
- Wenjin Zhao analyzed the data, authored or reviewed drafts of the paper, and approved the final draft.
- Jianhua Wang analyzed the data, prepared figures and/or tables, and approved the final draft.
- Lijian Peng performed the experiments, authored or reviewed drafts of the paper, and approved the final draft.
- Min Zhu conceived and designed the experiments, performed the experiments, analyzed the data, prepared figures and/or tables, authored or reviewed drafts of the paper, and approved the final draft.

## Data Availability

Data is available in the Admorph database's fish repository:

*Nostolepis amplifica*

IVPP V26832.1–IVPP V26832.41

DOI:10.12112/F.1320 –10.12112/F.1360

*Nostolepis digitus* sp. nov.

IVPP V26840.1–IVPP V26840.25

DOI:10.12112/F.1361 –10.12112/F.1385

*Nostolepis qujingensis* sp. nov.

IVPP V26838.1–IVPP V26838.16

DOI: 10.12112/F.1386–10.12112/F.1401

*Nostolepis striata*

IVPP V26830.1–IVPP V26830.68

DOI: 10.12112/F.1402–10.12112/F.1469

*Nostolepis consueta*

IVPP V26834.1–IVPP V26834.23

DOI:10.12112/F.1470–10.12112/F.1492

Investigated specimens were assigned accession numbers (IVPP V26830–IVPP V26840) and deposited at the Institute of Vertebrate Paleontology and Paleoanthropology (IVPP), Chinese Academy of Sciences, Beijing.

### New Species Registration

The following information was supplied regarding the registration of a newly described species:

Publication LSID: urn:lsid:zoobank.org:pub:C3957E52-DD5E-438E-BCD2-515B1611D9C2

Nostolepis qujingensis sp. nov. LSID: urn:lsid:zoobank.org:act:8AED8ED1-156A-4E2A-B274-B8E6D5319045

Nostolepis digitus sp. nov. LSID: urn:lsid:zoobank.org:act:40FFF454-861A-4FF4-82EC-9C3F455AAEE3

### Supplemental Information

Supplemental information for this article can be found online at http://dx.doi.org/10.7717/peerj.11093#supplemental-information.

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
