# Peer review of "Nostolepis scale remains (stem Chondrichthyes) from the Lower Devonian of Qujing, Yunnan, China"

_PeerJ, doi:10.7717/peerj.11093_

## Round 0.1 · original submission · Minor Revisions

Two reviewers are pleased with the MS and only minor proofreading corrections along with some other key details and better justification/reconsideration of one identification are needed. Please respond to the latter, more major points individually in your Response. Thank you!

·

Basic reporting

Overall, this is a welcome addition to the world of Palaeozoic microvertebrate studies. The biogeographic and biostratigraphical value of such remains has been proven for circum-Arctic regions by many studies over the last half-century, but the distribution of 'acanthodian' scale-based taxa outside Europe and Russia is relatively poorly known. Descriptions of Chinese assemblages help fill in the gaps in our knowledge.

Some of the English needs improvement, I have made suggestions on the annotated pdf.

Sufficient details of context and background have been provided. Literature references are comprehensive and relevant, although some could possibly be trimmed. I have made a few corrections to the reference list, and suggested where some references could be removed.

The article is professionally structured. Perhaps more figures of scale histology for the new species could be added. Figure 7 should be corrected as per my comments annotated to the figure pdf.

Experimental design

Zoobank ID details need to be added to the Materials and Methods section as per PeerJ instructions.

Validity of the findings

I am not convinced by the identification of one of the species, Nostolepis decora. As remarked in the annotated pdf, I wonder if the scales assigned to this species should rather be assigned to Hanilepis wangi. I realize that the type material for that species is older than the assemblage from the Xitun Fm described here, but to me the purported N decora scales are more similar morphologically and histologically to those of H wangi. If the authors agree about this assessment, they would need to amend the title and abstract - or rename Hanilepis wangi as Nostolepis wangi??

·

Basic reporting

No comment in general

As far as I can see, the article is written in clear, unambiguous, and technically correct English. I’ve attached a .pdf file, in which a couple of places are highlighted, where some changes might be done. However I am not a native English speaker, so I guess they are most probrbly unnecessary. The article provides sufficient coverage of Nostolepis taxonomic diversity, morphology, histology and paleogeography, and gives enough contextual information for the presented new data. All figures are of good quality, relevant to the topic, and references to them are given in the text. The article is ‘self-contained’.

Experimental design

No comment

The research theme is clearly defined.
The article describes six Nostolepis species from the Lochkovian of Qujing, eastern Yunnan, including 2 new species. It contributes to closing the gap in knowledge about paleogeographic distribution of Nostolepis and acanthodian diversity in the Early Devonian of South China.
It seems to me that taxonomic descriptions are rigorous enough: both morphology and histology of the scales are described in details and comparisons with the most similar known Nostolepis species are provided.
Maybe more data could be given about the samples, from which the scales were extracted and counts of the scales, but its not obligatory, I think. Methods of the study are described well enough.

Validity of the findings

No comment in general


The described findings are novel and could be of interest not only for paleoichyologists, but also for anyone interested in the large-scale Devonian stratigraphic correlation. For example, it has implications for correlation between South China and Baltica blocks. The commonly needed data for such type of studies are provided. The photographs of morphological features of the scales in different perspectives as well as the photographs of thin sections illustrating the scales’ histology are given for each of the described species.

Additional comments

Overall I find that the article provides important new knowledge on acanthodian diversity and must be published.
It could be of interest to see a map with localities, were the samples, from which the scales were obtained were collected and stratigraphic columns, but it seems to me that it is not obligatory.
I’ve attached a .pdf file, in which a couple of places are highlighted, where some changes might be done language-wise. However I am not a native English speaker, so I guess they are most probrbly not necessary.

---

## Round 0.2 · accepted · Accept

You have attended well to the reviewers' suggestions and the MS has nicely improved. The caution in the ID of N. decora seems to be a good move, in response to Reviewer 1. I am happy to recommend the paper for acceptance-- no further review is warranted.